# The Immediate and Lasting Effect of Emotion Regulation in Adolescents: An ERP Study

**DOI:** 10.3390/ijerph181910242

**Published:** 2021-09-29

**Authors:** Meng Yang, Xinmei Deng, Sieun An

**Affiliations:** 1School of Psychology, Shenzhen University, Shenzhen 518060, China; 1900482025@email.szu.edu.cn; 2Faculty of Psychology and Political Science, Eastern New Mexico University, Portales, NM 88130, USA; sieun.an@enmu.edu

**Keywords:** cognitive reappraisal, adolescents, lasting effect, emotion regulation, ERPs

## Abstract

The immediate effect is an important index of the outcomes of emotion regulation. However, in daily life, whether the effect of emotion regulation lasts and the lasting mechanism have been examined less. The present research focused on the relationships between the immediate and lasting effect of the emotion regulation of adolescents. Electroencephalogram (EEG) was recorded from 51 adolescents (31 boys and 20 girls, Mage = 12.82) during online emotion regulation using the Reactivity and Regulation-Image Task (phase 1) and re-presentation of emotional stimuli after a period of time (phase 2). Event-related potentials (ERPs) related to emotion regulation, such as N2, P3, and the late positive potential (LPP), were examined in the two phases. The results showed that: (1) In both of the two phases, in negative emotion conditions, the amplitudes of P3 and LPP 300–600 of no-regulation conditions were significantly higher than those in reappraisal conditions. However, there was no significant difference under neutral conditions; (2) The amplitudes of P3, N2, and LPP 300–600 during emotion regulation in phase 1 positively predicted the amplitudes of P3, N2, and LPP300–600 in phase 2 in different experimental conditions. Results from the regression analysis implied that the immediate effect of online emotion regulation may predict the lasting effect when adolescents face the same emotions again. In addition, our findings provide neurological evidence that the use of cognitive reappraisal could effectively help adolescents to reduce the recruitment of cognitive resources when they regulate negative emotions and when they face those negative emotions again.

## 1. Introduction

With the onset of puberty, individuals experience many changes in their physical, mental, and social lives, and enter an important period of emotional development. In an ocean of emotional storms, the ability to regulate emotions is particularly important for adolescents [1,2], but few studies have focused on this issue [3,4]. The present study focuses on the immediate and lasting effects of adolescents’ emotional regulation, and attempts to test and provide neurological evidence for this. We instructed adolescents to use a well-recognized, effective, and healthy strategy—cognitive reappraisal [5]—as a regulation strategy, and used event-related potentials (ERPs) to examine the effect of adolescents’ emotion regulation during an online voluntary emotional regulation phase and during a re-presentation phase after a period of time. We explored the impact of the immediate effects on the lasting effects for the first time. We hope that this can further our understanding of the staged characteristics of adolescents’ emotional regulation.

### 1.1. Emotion Regulation and Cognitive Reappraisal

Emotion regulation is the process by which individuals exert influence on the experience and expression of emotions [6]. Effective emotional regulation could help reduce negative emotions and enhance well-being, especially in the context of COVID-19 [7]. Individuals in this special period of puberty have a particularly high need for emotional regulation [1]. During adolescence, the development of an individual’s emotional regulation strategy undergoes complex changes [8] and is easily troubled by emotional dysregulation [9]. Effective emotion regulation strategies may not only help adolescents to respond adaptively to their emotional turbulence but also prevent dysfunctional regulation in later life and reduce the risk of psychological and physiological diseases related to emotions [2,10,11]. For example, a large number of studies have shown that reappraisal is one of the most flexible, effective, and adaptive regulatory strategies to modulate subjective emotional experience, facial expression, and psychophysiological arousal [12,13,14]. Reappraisal refers to the regulatory strategies that involve reinterpreting the meaning of the emotionally evocative stimuli, which employs knowledge as an emotion-generating system by changing cognitions about an emotional situation [15]. It is therefore of importance to further explore the effects of some easily deployed and available emotion regulation strategies (such as cognitive reappraisal) in adolescents.

### 1.2. The Immediate and Lasting Effect of Emotion Regulation

Since many emotional challenges in real life occur repeatedly, the longevity of emotion regulation over time is important. Because emotions are a multicomponential process that unfolds over time [16], emotion regulation involves changes in emotional dynamics such as duration, magnitude, and shape, and it is also a process that unfolds over time [17]. Research on the process of emotion regulation should not only focus on the effect during the voluntary emotion regulation process (i.e., immediate effect) but also the effect beyond the regulation period itself after the termination of voluntary emotion regulation, which we refer to as the lasting effect. Although a large number of studies have focused on the role of emotion regulation at the moment when an emotional event occurs, there is still a lack of attention on its lasting effects [3]. Previous experimental studies indicated that the effects of voluntary emotion regulation extend at least ten minutes beyond the period of active regulation [18]. Several subsequent studies found that although different emotion regulation strategies may play different roles in different stages of emotion regulation, only cognitive reappraisal appears to have a lasting effect [19,20]. This may be related to the stronger activation of the prefrontal regions (PFC) and amygdala [19,21]—that is, the depth of cognitive processing may affect the performance of lasting effects. Of course, the lasting change of cognitive reappraisal to the emotional response may be produced through different routes; studies have found that lasting effects on the amygdala response occurred even in the absence of continued prefrontal control [4]. Actually, the dynamics perspective could play an important role when exploring the process of emotional regulation, and temporal patterns of emotional responding provide a window into emotion regulatory processes [17]. However, there are currently few studies that have combined the immediate and lasting effects of emotion regulation to explore the relationship between the two.

From the perspective of age development, an fMRI study by Silvers and colleagues [21] found that during the active regulation process, older individuals showed a greater reduction in negative affect and inverse rostrolateral prefrontal–amygdala connectivity, and during the re-presentation process, they continued to show lasting reductions in the amygdala response mediated by the rostrolateral PFC. This study suggested that adults may have better long-term emotion regulation than adolescents due to better cognitive control. As to the form and neural mechanism of this lasting effect in adolescents, there is still a lack of research, and research combined with ERPs is even rarer.

### 1.3. ERP Components Related to Emotion Regulation

Due to the extremely high time resolution [22], which allows the time course of individuals’ brain electrophysiological activities to be clearly displayed, time-related potentials have been widely used in research related to emotion regulation. ERP components could reflect the characteristics of individuals’ cognitive processing during emotional arousal. Many studies have suggested that ERP indices of emotion regulation are related to modulations in attentional regulatory processes (e.g., P1, N2, P3, and early LPP components related to salience detection, stimulus-driven orientation, and attentional control) and cognitive regulatory processes (e.g., late LPPs related to cognitive control, meaning evaluation, and response modulation) [23,24,25,26].

The N2 component is a negative deflection observed at the fronto-central electrodes peaking between 200 and 350 ms after the presentation of a stimulus; it was often considered as an early neural marker of executive function and cognitive control [23,27]. Research on children and adolescents has also revealed this [28,29]. Moreover, N2 may also be sensitive to emotion-related information [30]. For example, a study conducted by Zhang and Lu [31] that combined the GO/NOGO paradigm found that GO-N2 reflects top-down attention to emotions and that these can be used as electrophysiological indices for automatic emotion regulation. The P3 component (or P3b) is a rather broad positive deflection observed at parietal electrodes peaking between 300 and 380 ms following stimulus presentation. The P3 appeared to reflect the allocation of capacity-limited resources toward motivationally salient environmental stimuli [22], which was related to individuals’ top-down processing. Some researchers regard this kind of allocation as a follow-up to initial efforts at self-regulation tapped by the N2 [24]; thus, P3 may reflect the second stage of emotion regulation.

The late positive potential (LPP) is evident as a broad superior posterior positive signal that occurs approximately 300 to 400 ms after stimulus onset [22]. In most studies of LPP, researchers divided this positive slow wave into different time windows and examined the average amplitude of the LPP within the relatively long time window [22,26,32]. The LPP reflects the facilitated attention to emotional stimuli; that is, it is not only sensitive to emotional stimulation itself but its time course is also an indicator of the emotion regulation process [33]. Different time windows of LPP are differentially sensitive to emotion regulation, and the psychological implications of their responses are also different. The research of MacNamara and colleagues [34] combined picture valence with a description framework, and found that the early time window of LPP was more sensitive to internal factors related to emotional meaning (i.e., picture valence), while the late LPP time window reflected the deeper processing related to the evaluation of emotional meaning (i.e., description frame). At the period of the late time window of LPP, the change in potential amplitude has changed from the increase in intentional attention caused by emotion perception to the intentional redistribution of attention caused by the use of regulation strategies [35].

As far as children and adolescents are concerned, researches have also found that LPP is an effective tool for studying the process and effects of emotion regulation, as well as age development differences [25,36,37], because its characteristics may be able to reflect the staged differences in the brain function and cognitive development of adolescent individuals. Many studies have suggested that when participants of different ages use reappraisal, there are differences in performance in different time windows of LPP [25,38]. Studies have found that there are differences in the amplitudes and time courses of LPP between adolescents and adults in the process of emotional regulation [36]. So far, however, compared with adults and children, there is still a lack of LPP studies on emotion regulation in adolescents.

### 1.4. Adolescents and Emotion Regulation

Since the transition to adolescence is accompanied by physical, psychological, and social changes, and these changes will trigger fresh emotional experiences [39], adolescence is a significant period for individual socio-emotional development. Adolescents experience frequent and fluctuating negative emotions in daily life, encounter unstable interpersonal relationships, are more vulnerable to engage in risk-taking behaviors, and are at greater risk of affective issues marked by emotional dysfunction [40,41]. These issues are thought to derive from adolescents’ still developing emotion regulation abilities and cognitive control abilities, which are underpinned by immaturity in adolescents’ social brains [42]. During adolescence, the limbic system of the human brain undergoes major changes, which may affect self-control, emotions, and decision making. At the same time, myelin synthesis in the frontal lobe related to cognitive processes is also increasing [43]. The advanced cognitive execution and social processes required for emotion regulation and corresponding brain regions are undergoing development during this period [44,45]. Examining adolescents’ emotion regulation from a neurodevelopmental perspective is of great significance. 

### 1.5. The Present Study

The present study explored the immediate and lasting effects of emotion regulation of adolescents in the context of cognitive reappraisal and no-regulation of emotional pictures. As mentioned above, while adolescents undergo significant development of emotion regulation, we still know little about the characteristics and neural relevance of emotion regulation in adolescents. We focused on the cognitive reappraisal strategy, and not only explored the variability in the N2, P3, and LPP components of ERP in the process of online emotion regulation (Phase 1) and re-presentation (Phase 2) in adolescents but also examined the relationship between the immediate and lasting effects of emotion regulation embodied in these two phases.

There were three hypotheses: (1) The valence of pictures and the regulation strategy used will influence the participant’s self-report ratings. Specifically, there will be greater self-reported subjective emotional intensity under the condition of negative valence and no-regulation strategies. (2) Works based on healthy adults have suggested that reappraisal of emotional stimuli could change the way they respond when re-exposed to stimuli later [3,4]. Therefore, we predicted that the effect of this strategy will be reflected in the corresponding ERP components regardless of the online regulation or re-presentation phase. Compared with neutral pictures, the amplitude of N2 will be larger when watching unpleasant negative pictures in both of the two phases. For the P3 component, differences in regulation strategies will appear, and there will not be much difference in performance in the two phases. Since adolescents’ cognitive control abilities are not yet well developed, recent work showed that the immediate and lasting effects of adolescents’ emotional regulation were relatively poor [21]. In view of this, for LPPs, we predict that valence and the use of strategies will have different effects on the temporal distribution of LPPs in the two phases (more precisely, the interaction of valence and the use of the strategy may appear in the early time window of LPP rather than the late time window). (3) The immediate effects of the first phase will positively predict the lasting effects of the second phase in N2, P3, and LPP early components. 

## 2. Methods

### 2.1. Participants

An a priori power analysis using the G*power software [46] revealed that a minimum of 46 participants would be required to detect a moderate effect size (0.4) in a correlational analysis (two-tails) with 80% power. This sample size is consistent with typical ERP studies [25,47]. It may be necessary to recruit more participants, given that some participants’ EEG data may be removed for reasons such as excessive movements. Thus, we recruited fifty-one Chinese early adolescents (31 males and 20 females, *M*_age_ = 12.82, *SD* = 1.36) who were each paid RMB 100 (equivalent to USD 16) for their participation. Participants were recruited via fliers that invited healthy volunteers from three junior middle schools to participate in a study of emotions. 

All participants resided in urban communities in China. Of the sample, 33.33% of the adolescents had no siblings, whereas others had one or more siblings. Approximately 78.43% of the fathers and 74.51% of the mothers had received a college education or above, and the other parents had received a high school or lower level of education. None of the participants in the present study had been trained in the mindfulness-based stress reduction technique or other mindfulness techniques before. All participants were right-handed and had normal or corrected-to-normal vision, and all were in good neurological and psychiatric condition. No participant had a history of neurological or psychiatric disorders, as determined by self- and/or parental report. Informed consent was obtained from the participants and their parents before commencing the study, and the participants were completely debriefed after the experiment. The research protocol was reviewed and approved by the Institutional Review Board of Shenzhen University. All procedures performed in studies involving human participants were conducted in accordance with the ethical standards of the institutional and/or national research committee and with the 1964 Declaration of Helsinki and its later amendments or comparable ethical standards.

### 2.2. Experimental Procedure

#### 2.2.1. Phase 1: Online Emotion Regulation (Immediate Effects of Emotion Regulation)

After the participants provided consent, they reported their demographic information. Next, after electroencephalograph (EEG) sensors were attached to the participants, they completed the Reactivity and Regulation-Image (REAR-I) Task. This task has been shown to successfully assess both emotional reactivity and emotion regulation over a broad age range in several previous studies [48,49,50]. Participants were tested individually in the room during the task. During the orientation section before the formal experiment, the participants received instructions for the task procedures adapted from the studies by Ochsner with colleagues [48] and Moser with colleagues [49]. They were told that in the subsequent experiment, they would view several pictures that might arouse different emotions, and they would need to regulate their emotions according to two possible instructions (reappraising emotions, and naturally reacting in response to the pictures). Then, the researcher provided each participant a list of samples of the two types of regulation and thoroughly defined reappraisal and no-regulation. The researcher was the author in this study who was experienced in the administration of the REAR-I task. For the look trials, the participants were asked to view the pictures and respond naturally. For the reappraisal trials, the participants were instructed to reappraise in a positive way to reduce their negative emotions, such as imagining that the situation had a better ending. The instructions were as detailed as necessary for the participants to understand them. As a primary manipulation assessment, the researcher asked the participants how they responded to the task under different experimental conditions to determine whether the participants understood the instructions. The responses from the participants indicated that they all understood the instructions and were able to regulate their emotions according to the instructions. Then, the formal experiment began. 

An experimental trial began with an instruction stating which emotional regulatory strategy to use (“reappraisal”/“look”) that was presented for one second. One negative or neutral picture was then shown for four seconds. The participants were told to view the picture and to regulate their emotional reaction to the picture using the instructed strategy. The continuous electroencephalogram was recorded during the online emotion regulation. Next, the participants rated their current intensity of emotion on a 9-point scale (i.e., How strongly do you feel after viewing the picture? 1 = very weak emotion to 9 = very strong emotion) by pressing a button. The inter-trial interval was 2 s. 

As to the experimental stimuli, eighty pictures were selected from the Chinese Affective Picture System (CAPS [51]): 40 negative (valence: *M* = 2.72, *SD* = 0.37; arousal: *M* = 5.40, *SD* = 0.38) and 40 neutral (valence: *M* = 5.39, *SD* = 0.34; arousal: *M* = 3.75, *SD* = 0.52) images. T-tests showed significant differences in the valence and arousal ratings between these two types of pictures. The negative pictures were more arousing and less pleasant than the neutral pictures (both *p*-values < 0.001). The pictures were age-appropriate for adolescents. The negative picture set included unpleasant social situations and frightening animals, and the neutral pictures depicted subjects of items such as household objects. The pictures (330 × 340 pixels) were presented in color on a 19-in monitor that occupied approximately 35° of the visual angle horizontally and vertically.

Four sessions and 160 experimental trials were included in the overall task, with 40 trials in each session. Eighty pictures were used in the task and were divided equally into four sessions, each of which included 20 negative and 20 neutral pictures. Each picture was repeated one time for each of the two regulatory instructions. Each session consisted of 4 experimental blocks, and the 4 experimental blocks were randomly shuffled within one session. The order of the trials was randomly presented within each block. The task was administered using E-Prime software. The cursors were positioned randomly, and the scores were recorded using a keyboard. An experimental session of 20–25 min in phase 1 was required for each participant (Figure 1).

#### 2.2.2. Phase 2: Re-presentation of Emotional Stimuli (Lasting Effects of Emotion Regulation)

Five minutes after finishing the 25-min REAR-I task, participants were presented with all 40 pictures that they had previously seen during the REAR-I task to examine the lasting effect of emotion regulation in phase 1. Re-presentation of emotional stimuli was designed to examine whether prior emotion regulation influenced the speed of responses to emotional stimuli. Participants were not told they would be re-presented with the stimuli they had previously seen in the REAR-I task. In phase 2, the 80 pictures were randomly re-presented twice, with a total of 160 trials. At the beginning of each trial, a fixation point was presented for 500 ms, followed by a stimulus photo that was presented for 2000 ms. The continuous electroencephalogram was recorded during the stimuli presented. Next, the participants were asked to rate their current intensity of emotion on a 9-point scale (i.e., How strongly do you feel after viewing the picture? 1 = very weak emotion to 9 = very strong emotion) by pressing a button. The inter-trial interval was 1000 ms. The protocol was adopted and modified from Silvers and colleagues’ study [21].

### 2.3. Psychophysiological Recordings and Data Reduction

To examine the instant and lasting effects of emotion regulation, continuous electroencephalogram were recorded in both phase 1 and phase 2 with a 64-channel amplifier (BrainAmp, Brain Products, Germany) based on the 10/20 system, with two electrodes placed on the left and right mastoids. In phase 1, the continuous electroencephalogram was recorded during the online emotion regulation. In phase 2, the continuous electroencephalogram was recorded during the stimuli presented. The electroencephalogram was sampled at 500 Hz, and impedance was maintained at less than 5 kΩ. The data were referenced offline to the averaged mastoid references and bandpass filtered from 0.5 Hz to 30 Hz [52]. Eye movements and blink artifacts were corrected using the independent component analysis (ICA) algorithm implemented in Brain Vision Analyzer 2.0 (“Brain Products”, Germany). The data were segmented in epochs from 200 ms before the onset of stimuli until 1500 ms after the stimulus onset. In phase 1 and phase 2, the ERPs were constructed by separately averaging them according to the four experimental conditions (negative no-regulation, negative reappraisal, neutral no-regulation, and neutral reappraisal). For each ERP average, the average activity in the 200 ms before stimulus onset served as the baseline. Trials with artifacts exceeding ±80 μV were excluded from further analysis. In phase 1, the mean number of valid epochs averaged per condition was 30.12 (75.29%) for negative no-regulation, 30.98 (77.45%) for negative reappraisal, 31.43 (78.58%) for neutral no-regulation, and 30.71 (76.76%) for neutral reappraisal. In phase 2, the mean number of valid epochs averaged per condition was 32.22 (80.54%) for negative no-regulation, 32.14 (80.34%) for negative reappraisal, 32.20 (80.49%) for neutral no-regulation, and 32.61 (81.52%) for neutral reappraisal.

Based on the existing literature and a visual inspection, in phase 1, N2 was evaluated at Cz as the largest peak voltage between 220 and 290 ms after stimulus onset, and in phase 2, N2 was evaluated at Cz as the largest peak voltage between 220 and 270 ms after stimulus onset [53]. P3 was evaluated at Pz as the largest peak voltage between 300 and 380 ms after stimulus onset in both phase 1 and 2 [54]. LPP is a long-lasting positive signal that occurs approximately 300 to 400 ms after stimulus onset [22]. In most studies of LPP, researchers divided this positive slow wave into different time windows and examined the average amplitude of the LPP within the relatively long time window [22,25,32]. In the present study, the LPP was defined as the average activity at Pz [36] in three time windows after stimulus onset: LPP 300–600 ms (early window), LPP 600–1000 ms (middle window), and LPP1000–1500 ms (late window).

### 2.4. Behavioral and ERP Data Analysis

In both phase 1 and phase 2, behavioral ratings of emotional experience intensity under the different experimental conditions were examined using a 2 (valence: neutral vs. negative) × 2 (strategy: no-regulation vs. reappraisal) repeated measures ANOVA. 

To examine the immediate effects of emotion regulation, 2 (valence: neutral vs. negative) × 2 (strategy: no-regulation vs. reappraisal) repeated measures ANOVAs were conducted separately for the N2, P3, and LPPs in phase 1 (online emotion regulation). To examine the lasting effects of emotion regulation, 2 (valence: neutral vs. negative) × 2 (strategy: no-regulation vs. reappraisal) repeated measures ANOVAs were conducted separately for the N2, P3, and LPPs in phase 2 (re-presentation of emotional stimuli). Furthermore, to examine how immediate effects related with lasting effects of emotion regulation, correlational analyses were conducted for the ERPs (N2, P3, and LPPs) between phase 1 and phase 2 in different experimental conditions.

Bonferroni correction was used for multiple post hoc comparisons, and the p-value reported below was corrected. The behavioral ratings, N2, P3, and LPPs were statistically evaluated using SPSS 20.0. The Greenhouse–Geisser correction was applied to *p*-values associated with multiple-*df* comparisons.

Family structure (single child family) and parental educational levels were used as indicators of the social economic status of the participants. Family structure and parental educational levels were used as covariates to ensure that the regulation-related decreases in ERPs were not influenced by the social economic status of the participants. Results of the repeated measures ANOVA indicated that the main effects of the relevant variables on the ERPs were not significant in both phase 1 and phase 2. Therefore, these variables were not included in subsequent analyses. Details of the repeated measures ANOVA are presented in Appendix A.

## 3. Results

### 3.1. Behavioral Results

#### 3.1.1. Phase 1: Online Emotion Regulation

The main effect of valence was significant, *F* (1, 50) = 72.62, *p <* 0.001, η_p_^2^ = 0.59. The arousal ratings of the negative stimuli were higher than the neutral stimuli (*p* < 0.001). The main effect of the regulation strategy was significant, *F* (1, 50) = 10.40, *p* = 0.002, η_p_^2^ = 0.17. The arousal ratings of no-regulation were higher than reappraisal, *p* = 0.002. The interaction between valence and the regulation strategy was also significant, *F* (1, 50) = 10.84, *p =* 0.002, η_p_^2^ = 0.18. For the negative stimuli, the arousal ratings of no-regulation were higher than reappraisal (*p* = 0.001). For neutral stimuli, there was no significant difference between the arousal ratings of no-regulation and reappraisal (*p* = 0.108).

#### 3.1.2. Phase 2: Re-presentation of Emotional Stimuli

The main effect of valence was significant, *F* (1, 50) = 57.90, *p <* 0.001, η_p_^2^ = 0.54. The arousal ratings of the negative stimuli were higher than the neutral stimuli (*p* < 0.001). The main effect of the regulation strategy was not significant, *F* (1, 50) = 0.05, *p* = 0.828, η_p_^2^ = 0.00. The interaction between valence and the regulation strategy was not significant, *F* (1, 50) = 0.22, *p =* 0.645, η_p_^2^ = 0.00. Descriptive statistics of the behavioral ratings in both of the two phases were listed in Table 1.

### 3.2. Neural Results

The descriptive statistical results of the average amplitudes of each of the ERP components under different conditions in the two phases are shown in Table 2 and Figure 2.

#### 3.2.1. Phase 1: Online Emotion Regulation (Immediate Effects of Emotion Regulation)

In order to examine the immediate effects of emotion regulation, 2 (valence: neutral vs. negative) × 2 (strategy: no-regulation vs. reappraisal) repeated measures ANOVAs were conducted separately for the N2, P3, and LPPs in phase 1 (Online Emotion Regulation).

N2. The main effect of valence was significant, *F* (1, 50) = 57.09, *p <* 0.001, η_p_^2^ = 0.53. The amplitude of N2 was larger for the negative stimuli than for the neutral stimuli (*p <* 0.001). The main effect of the regulation strategy was not significant, *F* (1, 50) = 0.95, *p* = 0.334, η_p_^2^ = 0.02. There was no significant interaction between emotional valence and the regulation strategy, *F* (1, 50) = 0.00, *p* = 0.966, η_p_^2^ = 0.00.

P3. There was a significant main effect of valence, *F* (1, 50) = 28.04, *p* < 0.001, η_p_^2^ = 0.36. The amplitude of P3 was larger for the negative stimuli than for the neutral stimuli (*p* < 0.001). The main effect of the regulation strategy was significant, *F* (1, 50) = 16.73, *p <* 0.001, η_p_^2^ = 0.25. The amplitude of P3 was larger for no-regulation than for reappraisal (*p <* 0.001). The interaction between valence and the regulation strategy was significant, *F* (1, 50) = 16.42, *p <* 0.001, η_p_^2^ = 0.25. For the negative stimuli, the P3 amplitudes of no-regulation were higher than reappraisal (*p* < 0.001). For neutral stimuli, there was no significant difference between the P3 amplitudes of no-regulation and reappraisal (*p* = 0.569).

LPP. For LPP 300–600 ms, the main effect of valence was significant, *F* (1, 50) = 20.33, *p <* 0.001, η_p_^2^ = 0.29. The amplitude of the LPP was larger for the negative stimuli than for the neutral stimuli (*p <* 0.001). The main effect of the regulation strategy was not significant, *F* (1, 50) = 2.03, *p =* 0.160, η_p_^2^ = 0.04. The interaction between valence and the regulation strategy was marginally significant, *F* (1, 50) = 3.91, *p =* 0.054, η_p_^2^ = 0.07. For the negative stimuli, the LPP amplitudes of no-regulation were higher than reappraisal (*p* = 0.026). For neutral stimuli, there was no significant difference between the LPP amplitudes of no-regulation and reappraisal (*p* = 0.809).

For LPP 600–1000 ms, there was a significant main effect of valence, *F* (1, 50) = 44.14, *p <* 0.001, η_p_^2^ = 0.47. The LPP amplitude was larger for the negative stimuli than for the neutral stimuli (*p* < 0.001). The main effect of the regulation strategy was not significant, *F* (1, 50) = 0.78, *p =* 0.382, η_p_^2^ = 0.02, and the interaction between valence and the regulation strategy was not significant, *F* (1, 50) = 1.06, *p =* 0.309, η_p_^2^ = 0.02.

For LPP 1000–1500 ms, there was a significant main effect of valence, *F* (1, 50) = 11.89, *p =* 0.001, η_p_^2^ = 0.19. The LPP amplitude was larger for the negative stimuli than for the neutral stimuli (*p* = 0.001). The main effect of the regulation strategy was not significant, *F* (1, 50) = 0.01, *p =* 0.940, η_p_^2^ = 0.00, and the interaction between valence and the regulation strategy was not significant, *F* (1, 50) = 0.30, *p =* 0.587, η_p_^2^ = 0.01.

#### 3.2.2. Phase 2: Re-presentation of Emotional Stimuli (Lasting Effects of Emotion Regulation)

In order to examine the lasting effects of emotion regulation, 2 (valence: neutral vs. negative) × 2 (strategy: no-regulation vs. reappraisal) repeated measures ANOVAs were conducted separately for the N2, P3, and LPPs in phase 2 (Re-presentation of Emotional Stimuli).

N2. The main effect of valence was significant, *F* (1, 50) = 26.41, *p <* 0.001, η_p_^2^ = 0.35. The amplitude of the N2 was larger for the negative stimuli than for the neutral stimuli (*p <* 0.001). The main effect of the regulation strategy was significant, *F* (1, 50) = 6.12, *p =* 0.017, η_p_^2^ = 0.11. The amplitude of the N2 was larger for the no-regulation than for reappraisal (*p =* 0.017). The interaction between valence and the regulation strategy was not significant, *F* (1, 50) = 2.96, *p =* 0.092, η_p_^2^ = 0.06.

P3. There was a significant main effect of valence, *F* (1, 50) = 7.89, *p* = 0.007, η_p_^2^ = 0.14. The amplitude of P3 was larger for the negative stimuli than for the neutral stimuli (*p* = 0.007). The main effect of the regulation strategy was significant, *F* (1, 50) = 4.27, *p =* 0.044, η_p_^2^ = 0.08. The amplitude of P3 was larger for no-regulation than for reappraisal (*p =* 0.044). The interaction between valence and the regulation strategy was significant, *F* (1, 50) = 12.72 *p =* 0.001, η_p_^2^ = 0.20. For the negative stimuli, the P3 amplitudes of no-regulation were higher than reappraisal (*p* < 0.001). For neutral stimuli, there was no significant difference between the P3 amplitudes of no-regulation and reappraisal (*p* = 0.518). 

LPP. For LPP 300–600 ms, the main effect of valence was significant, *F* (1, 50) = 25.79, *p <* 0.001, η_p_^2^ = 0.34. The amplitude of the LPP was larger for the negative stimuli than for the neutral stimuli (*p <* 0.001). The main effect of the regulation strategy was not significant, *F* (1, 50) = 1.03, *p =* 0.316, η_p_^2^ = 0.02. The interaction between valence and the regulation strategy was significant, *F* (1, 50) = 10.97, *p =* 0.002, η_p_^2^ = 0.18. For the negative stimuli, the LPP amplitudes of no-regulation were higher than reappraisal (*p* = 0.005). For neutral stimuli, there was no significant difference between the LPP amplitudes of no-regulation and reappraisal (*p* = 0.181).

For LPP 600–1000 ms, there was a significant main effect of valence, *F* (1, 50) = 33.16, *p <* 0.001, η_p_^2^ = 0.40. The LPP amplitude was larger for the negative stimuli than for the neutral stimuli (*p* < 0.001). The main effect of the regulation strategy was not significant, *F* (1, 50) = 0.74, *p =* 0.393, η_p_^2^ = 0.02, and the interaction between valence and the regulation strategy was not significant, *F* (1, 50) = 1.24, *p =* 0.271, η_p_^2^ = 0.02.

For LPP 1000–1500 ms, the main effect of valence was significant, *F* (1, 50) = 5.86, *p =* 0.019, η_p_^2^ = 0.11. The main effect of the regulation strategy was not significant, *F* (1, 50) = 0.08, *p =* 0.786, η_p_^2^ = 0.00, and the interaction between valence and the regulation strategy was not significant, *F* (1, 50) = 0.15, *p =* 0.699, η_p_^2^ = 0.00.

The grand averaged waveforms and scalp distributions under different conditions were shown in Figure 3.

### 3.3. Relationships between Immediate and Lasting Effects of Emotion Regulation

Linear regressions were conducted to examine the relationship between the immediate and lasting effects of emotion regulation. As shown in Table 3, the immediate effects of emotion regulation in phase 1 positively predicted (*beta* = 0.45~0.78) the lasting effects of emotion regulation in phase 2 for P3, N2 and LPP 300–600 in different experimental conditions.

## 4. Discussion

Due to the physical, psychological, social, and other challenges that accompany adolescence [55], it is a period vulnerable to emotional dysregulation. The improvement in the developmental characteristics of emotional regulation in this period may be of great value for the prevention of psychopathology [2,9]. Recent work has suggested that adolescents are more likely to experience extreme and turbulent emotions, possibly due to immature cognitive control ability over emotion [21,56]. The present research builds on the previous research and explores the immediate and lasting effects of adolescents using cognitive reappraisal strategies to regulate emotions, and examines the predictive effect of the immediate effects on the lasting effects. Overall, the results suggest that a cognitive reappraisal strategy can effectively regulate adolescents’ negative emotions in phase 1 (online emotion regulation) and phase 2 (re-presentation), but this effect was only reflected in the ERP components (such as P3 and LPP early window) related to attentional regulatory processes; there was no significant effect on the late ERP components (LPP middle window and LPP late window) related to cognitive regulatory processes [25,34]. On the other hand, the immediate effect of phase 1 could positively predict the lasting effect of phase 2, but this prediction effect was only reflected in some relatively early ERP components (such as N2, P3, and LPP early window).

In both the online emotion regulation phase and the stimulus re-presentation phase, the ERP components were sensitive to the valence of the stimuli, and the ERP amplitudes induced by negative stimuli were larger than that of neutral stimuli, which also suggested the stimuli we chose had a strong degree of discrimination. Although in the online regulation phase N2 was not sensitive to the regulation strategy or the interaction between the strategy and valence, in the re-presentation stage we found that the main effect of the strategy was significant, which means that when adolescents were faced with the stimulus that was re-evaluated before, the N2 amplitude was smaller. As mentioned earlier, N2 may be related to emotional attention [23,27]. Such results indicated that cognitive reappraisal may help adolescents to reduce their intentional attention when facing the same emotional stimuli again, thereby reducing the cognitive resources that need to be recruited and reducing cognitive load. Whether in the online emotion regulation or re-presentation phase, we observed the main effect of valence, the strategy, and the interaction between valence and the strategy on the P3 component. P3 could reflect the allocation of cognitive resources triggered by individuals’ motivation, similar to N2, as it increases with higher cognitive control demands [22,24]. This result was in line with our expectations; it showed that the use of cognitive reappraisal could effectively help adolescents to reduce the cognitive resource load required when they adjust negative emotions and when they face negative emotions again, thereby reducing the impact of negative emotions.

Regarding LPPs, we found that only the early LPP component was sensitive to the effect of valence and the interaction between the strategy and valence; the middle and late components of LPP were only sensitive to valence. As was already known, LPP is a typical ERP component for studying the process of emotion regulation [33]. The variation trend of LPP reflects the time course of different nervous systems involved in emotion regulation processing [57]. For adolescents, in the LPP early time window, the amplitude of negative emotions under the condition of cognitive reappraisal was smaller than that under the condition of no-regulation, which indicated that the use of the cognitive reappraisal strategy could save adolescents from excessive use of neurological and cognitive resources when facing negative emotional challenges. However, correspondingly, the immaturity of adolescents’ brain function and cognitive development were also demonstrated by the result of their insensitivity to the effects of the strategy in the LPP middle and late time windows. As stated before, the LPP late time window reflected the deeper processing related to the evaluation of emotional meaning, suggesting it is an important indicator of cognitive resource use and effective cognitive regulation [32,36]. It can be concluded from our results that adolescents’ cognitive development was still inadequate, and the processing of emotional meaning was still at a relatively shallow stage. Their ability to process and evaluate deep emotional meaning was still not enough, and they were not yet in a position to make full use of cognitive reappraisal.

In terms of the results of the regression analysis, we found that the immediate effects of emotion regulation in phase 1 positively predicted the lasting effects in phase 2 in P3, N2, and LPP 300–600 in different experimental conditions. This result indicated to a certain extent that the better the effect of online emotional regulation the first time, the better the regulation effect would be when receiving the same emotional stimulation again later. However, such a positive prediction effect didn’t occur in the LPP middle and late time window. This result may well correspond to the previous analysis, that is to say, because the development of adolescents’ cognitive processing ability is not yet mature, the lasting effect of emotional regulation does not reflect on a deeper level. Since emotional abilities are still developing [8,44], adolescents at this stage could not conduct a deep evaluation of emotional meaning in the online emotional regulation phase, so the effect of deep processing was even more difficult to continue until the subsequent phase of re-presentation.

In addition, the self-reported analysis results are also worth mentioning. We found that in the process of online emotion regulation in phase 1, the participants’ subjective emotional arousal ratings were sensitive to the effects of valence, the strategy, and the interaction between valence and the strategy. However, in phase 2, when the emotional stimuli were presented again, the participants’ subjective arousal ratings were only sensitive to the effect of emotional valence. This result reinforces that, from a subjective point of view, when facing the previous emotional stimulus again, the participants may not have felt the difference in strategy use. On the one hand, this may be due to the weakening of the effect of cognitive reappraisal over time, which was somewhat analogous to the research results of MacNamara and colleagues [3]. This also suggests that the use of the cognitive reappraisal strategy for adolescents may not be as effective—at least in terms of subjective arousal ratings. However, from another perspective, although the use of the strategy didn’t have a significant impact on subjective ratings, it was reflected in the measurement of objective brain electrical activity, which may indicate that the effect of cognitive reappraisal was more reflected on the subconscious level.

## 5. Conclusions 

The findings of the current study suggest that cognitive reappraisal can help adolescents better regulate negative emotions, and can also exert a lasting effect when facing the same emotional challenge again. In addition, the immediate effect could positively predict the lasting effect to a certain extent. However, due to the immaturity of adolescents’ neurological and cognitive function development, the subjects could not fully utilize cognitive reappraisal. Specifically, the influence of the strategy didn’t appear in the LPP middle and late time windows related to deep-level meaning processing. There are still some limitations to be considered in the present study. First of all, our rest time is relatively short, and we tested short-term regulation rather than repetitive training or the accumulation of learning. The small number of exercises and learning time may make it difficult to change adolescents’ deep processing of emotional stimuli, which will influence our experimental results to a certain extent. Learning and mastering an emotion regulation strategy should be a gradual process, and the impact of long-term use of cognitive reappraisal strategy could be explored in the future. Second, our participants may still be young (*M*_age_ = 12.82, *SD* = 1.36), and the effect of using emotion regulation strategies may change throughout development. Therefore, future research should investigate the neurological and behavioral effects of using different emotion regulation strategies in samples of different ages. Finally, although we controlled the influence of family structure and parental education levels, for developing youth, there are still factors including family parenting styles, adolescents’ emotional state, and depression and anxiety characteristics that need to be considered.

## Figures and Tables

**Figure 1 ijerph-18-10242-f001:**
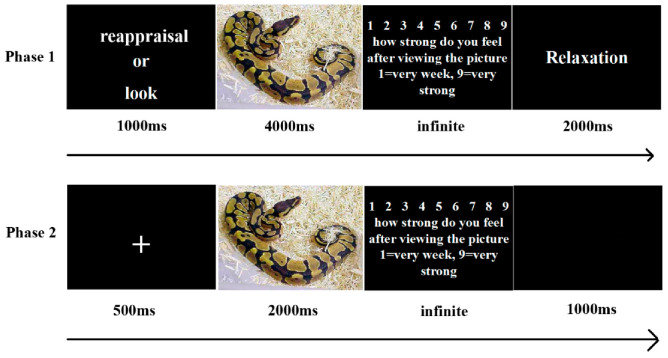
Procedure of the Study.

**Figure 2 ijerph-18-10242-f002:**
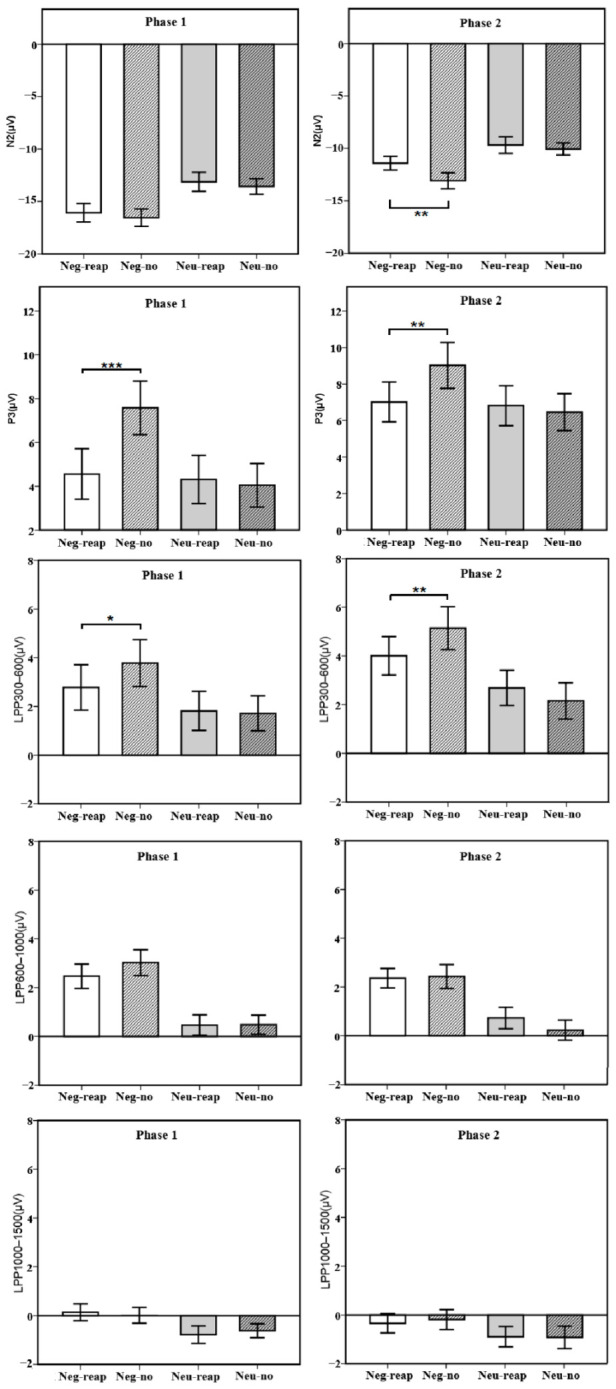
Average amplitudes of different components between different conditions in phase 1 and phase 2 (N2, P3, LPP300–600, LPP600–1000, and LPP1000–1500). * *p* < 0.05; ** *p* < 0.01; *** *p* < 0.001. Neg-reap = Negative reappraisal; Neg-no = Negative no-regulation; Neu-reap = Neutral reappraisal; Neu-no = Neutral no-regulation. Error Bars represent standard error.

**Figure 3 ijerph-18-10242-f003:**
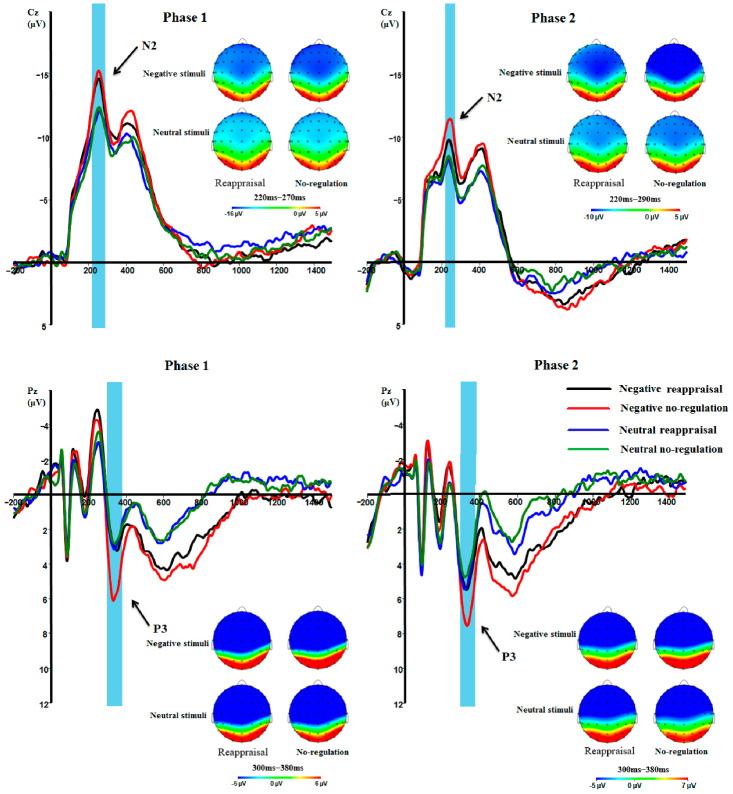
Stimulus-locked grand averaged waveforms and scalp distributions of mean activity between different conditions in Cz and Pz.

**Table 1 ijerph-18-10242-t001:** Descriptive Statistics for Behavioral Ratings.

Stimulus Type	Phase	Arousal (*M* ± *SD*)
Negative reappraisal	1	3.37 ± 1.85
	2	3.35 ± 2.11
Negative no-regulation	1	3.71 ± 2.02
	2	3.37 ± 2.18
Neutral reappraisal	1	1.86 ± 0.88
	2	1.88 ± 1.04
Neutral no-regulation	1	1.91 ± 0.88
	2	1.87 ± 1.06

Note. *N* = 51.

**Table 2 ijerph-18-10242-t002:** Average Amplitudes of Different Components between Different Conditions.

ERPs	Condition	Amplitude (*M* ± *SD*)	
		Phase 1	Phase 2
N2	Negative reappraisal	−16.08 ± 6.28	−11.40 ± 4.66
	Negative no-regulation	−16.56 ± 5.88	−13.09 ± 5.45
	Neutral reappraisal	−13.13 ± 6.46	−9.68 ± 5.71
	Neutral no-regulation	−13.57 ± 5.35	−10.06 ± 4.10
P3	Negative reappraisal	4.56 ± 8.26	7.01 ± 7.79
	Negative no-regulation	7.58 ± 8.74	9.02 ± 9.00
	Neutral reappraisal	4.31 ± 7.88	6.81 ± 7.82
	Neutral no-regulation	4.04 ± 7.09	6.46 ± 7.21
LPP 300–600 ms	Negative reappraisal	2.78 ± 6.68	4.00 ± 5.64
	Negative no-regulation	3.78 ± 6.88	5.14 ± 6.29
	Neutral reappraisal	1.81 ± 5.73	2.68 ± 5.18
	Neutral no-regulation	1.72 ± 5.13	2.15 ± 5.34
LPP 600–1000 ms	Negative reappraisal	2.47 ± 3.55	2.36 ± 2.83
	Negative no-regulation	3.02 ± 3.80	2.43 ± 3.49
	Neutral reappraisal	0.47 ± 2.97	0.73 ± 3.16
	Neutral no-regulation	0.48 ± 2.77	0.23 ± 2.96
LPP 1000–1500 ms	Negative reappraisal	0.13 ± 2.45	−0.34 ± 2.82
	Negative no-regulation	0.01 ± 2.31	−0.19 ± 2.91
	Neutral reappraisal	−0.78 ± 2.55	−0.89 ± 2.97
	Neutral no-regulation	−0.62 ± 2.04	−0.92 ± 3.28

Note. *N* = 51.

**Table 3 ijerph-18-10242-t003:** Regression Analyses of Immediate and Lasting Effects of Emotion Regulation.

ERPs	Condition	β	*t*	*R^2^*	*F*
N2	Negative reappraisal	0.54 ***	4.46	0.29	19.92 ***
	Negative no-regulation	0.54 ***	4.53	0.30	20.48 ***
	Neutral reappraisal	0.60 ***	5.24	0.40	27.49 ***
	Neutral no-regulation	0.45 **	3.48	0.20	12.09 **
P3	Negative reappraisal	0.78 ***	8.71	0.61	75.81 ***
	Negative no-regulation	0.78 ***	8.60	0.60	73.98 ***
	Neutral reappraisal	0.77 ***	8.52	0.60	72.59 ***
	Neutral no-regulation	0.69 ***	6.60	0.47	43.53 ***
LPP 300–600 ms	Negative reappraisal	0.73 ***	7.47	0.53	55.80 ***
	Negative no-regulation	0.70 ***	6.94	0.49	48.14 ***
	Neutral reappraisal	0.72 ***	7.35	0.52	54.02 ***
	Neutral no-regulation	0.60 ***	5.29	0.36	27.94 ***
LPP 600–1000 ms	Negative reappraisal	0.10	0.72	0.01	0.52
	Negative no-regulation	0.05	0.38	0.00	0.14
	Neutral reappraisal	0.20	1.43	0.04	2.05
	Neutral no-regulation	0.22	1.59	0.05	2.54
LPP 1000–1500 ms	Negative reappraisal	0.05	0.37	0.00	0.14
	Negative no-regulation	0.15	1.07	0.02	1.14
	Neutral reappraisal	0.16	1.11	0.03	1.23
	Neutral no-regulation	0.40 **	3.09	0.16	9.57 **

Note. *N* = 51. ** *p* < 0.01; *** *p* < 0.001. Predictors were the amplitudes of ERPs in the four experimental conditions in phase 1, and predicted variables were the amplitudes of the ERPs in the four experimental conditions in phase 2.

## Data Availability

The data presented in this study are available on request from the corresponding author.

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
