# Peer review of "The Immediate and Lasting Effect of Emotion Regulation in Adolescents: An ERP Study"

_ijerph, 2021, doi:10.3390/ijerph181910242_

Round 1
Reviewer 1 Report
This article presents an interesting topic on the relationships between the immediate and lasting effect of emotion regulation of adolescents .
It is executed at a good level, corresponds to the accepted standards, built design, goals and conclusions are correlated, a deep analysis of the literature and no doubt the reader's interest in the topic.
During the expertise a number of comments arose:
1. No date and consent number of the ethics committee
2. There are no abbreviations in the thesis EEG, ERPs, LPP -not clear .
3. You have the abbreviation repeated on line 36 and again on line 95 -ERPs
4. It is not clear what is meant in line 92 - PFC
5. Were the medications evaluated - did the children take anything?
6. Table 1 would be better if you moved it to one sheet.
7. I don't understand why in table 1 and 2 the number of participants is 51, and in the methods 46.
8. Figure 3 should be enlarged.
9. The list of references includes old sources more than 5 years old, review.
Reviewer 2 Report
I would like to congratulate the authors for proposing this strategy for emotional regulation in adolescents with promising results. Although the sample size is not very large, it is positive to assess 51 subjects neuropsychologically with such cumbersome tests.
I would strongly recommend updating the bibliography, given that the most current citation is from 2019 and the rest are more than 5 years old. It would be interesting to know emotionally what has happened during COVID-19 confinement in adolescents in general.
